# Research Progress on Mango Post-Harvest Ripening Physiology and the Regulatory Technologies

**DOI:** 10.3390/foods12010173

**Published:** 2022-12-30

**Authors:** Bangdi Liu, Qi Xin, Min Zhang, Jianhu Chen, Qingchen Lu, Xinqun Zhou, Xiangxin Li, Wanli Zhang, Wei Feng, Haisheng Pei, Jing Sun

**Affiliations:** 1Academy of Agricultural Planning and Engineering, Ministry of Agriculture and Rural Affairs, Beijing 100125, China; 2Key Laboratory of Agro-Products Primary Processing, Ministry of Agriculture and Rural Affairs, Beijing 100125, China; 3College of Life Science and Food Engineering, Hebei University of Engineering, Handan 056038, China; 4Institute of Apicultural Research, Chinese Academy of Agricultural Sciences, Beijing 100093, China; 5School of Food Science and Engineering, Hainan University, Haikou 570228, China

**Keywords:** postharvest mango, ripening, physiological changes, metabolic pathways, preservation technology

## Abstract

Mango (*Mangifera indica* L.) is an important tropical fruit with a delicate taste, pleasant aroma, and high nutritional value. In recent years, with the promotion of the rural revitalization strategy and the development of the poverty alleviation industry, China has gradually become an important mango producer. However, the short shelf life of mango fruit, the difficulty in regulating the postharvest quality, and the lack of preservation technology are the main problems that need to be solved in China‘s mango industry. In this paper, the physiological changes and mechanisms of mango during postharvest ripening were summarized, including sugar and acid changes, pigment synthesis and accumulation, and aroma formation and accumulation. The physical, chemical, and biological technologies (such as endogenous phytohormones, temperature, light, chemical preservatives, and edible coatings) commonly used in the regulation of mango postharvest ripening and their action principles were emphatically expounded. The shortcomings of the existing mango postharvest ripening regulation technology and physiological mechanism research were analyzed in order to provide a reference for the industrial application and development of mango postharvest.

## 1. Introduction

Mango (*Mangifera indica* L.) has a unique flavor and pleasant aroma and contains a variety of nutrients. It is highly popular among consumers and has gradually developed into the second most grown and consumed tropical fruit in the world [1]. According to the data from the Food and Agriculture Organization of the United Nations, the global production of mango is approximately 54.83 million tons in 2020, with the planted area exceeding 55 million hectares (Figure 1). The production areas of mango are mainly distributed in tropical regions with high temperatures and humidity and are relatively dispersed [2,3]. With the continuous expansion of the consumer market, the consumption form of mango has gradually transformed from local and peripheral sales to off-site and export sales modes. However, over extended periods of cold-chain transportation, spoilage and deterioration in the quality of mango caused by postharvest ripening and aging during storage and transportation often occur, resulting in high economic losses.

Currently, India is the largest mango-producing country [4]. China has gradually developed into the world’s second largest mango producer after completing its poverty alleviation campaign and promotion of the rural revitalization strategy [4]. In 2020, China’s mango cultivation area accounted for 349,000 hectares, with an output of 3.306 million tons (Figure 1) valued at approximately CNY 20.52 billion [2]. Mango cultivation in China is mainly distributed across Hainan, Guangxi, and Yunnan provinces. The mango industry has become key for the promotion of the rural revitalization strategy and consolidation of poverty eradication achievements in the central and western regions, economically less developed regions, and remote mountainous areas of China [5,6]. Research on mango preservation technologies can aid in reducing economic losses during storage and distribution.

Postharvest ripening is the process by which the fruit continues to shift from harvest maturity to edibility, thereby increasing the economic and consumption values of the fruit. With the development of the mango industry, marketing methods for raw mangoes are constantly evolving [7], and consumers’ expectations of fruit quality continue to increase [8]. The existing mango preservation and ripening technology has stagnated at the traditional method of low-temperature storage and transportation at 13 °C to delay ripening and storage in an enclosed warehouse (vehicle) after ethephon spraying to promote ripening [2]. This method has limited ability to control the postripening of mango, and it is easy for mangoes in the entire warehouse (entire vehicle) to rot due to rapid ripening and aging, causing considerable economic losses and biological pollution. Therefore, a review of the postharvest ripening of mango, its mechanisms, and the regulation technology can guide the mango industry and provide a theoretical basis for the development of new postharvest ripening delay or ripening technologies.

## 2. Physiological Changes in Mango during Ripening

There are several diverse varieties of mango. Well-ripened fruits have a distinctive flavor, a mix of sweet and sour, are soft and juicy, and accompanied by a pleasant aroma, which can be attributed to a series of physiological changes caused by the activation of chain reactions of molecules and metabolic pathways related to postharvest ripening [9]. As shown in Figure 2, these changes in mango mainly include enhanced respiration and ethylene release, changes in the cell wall microstructure and texture softening, gluconeogenesis, acid decomposition, pigment synthesis and accumulation, color change, and the synthesis and accumulation of characteristic aromatic compounds.

### 2.1. Changes in Respiration, Ethylene, and Energy Levels during Postharvest Ripening of Mango

Mango is a typical climacteric fruit. During the ripening process, a clear climacteric peak occurs, with a sharp increase in respiration intensity accompanied by the rapid synthesis of ethylene [2]. The organic matter of the fruit is oxidized and broken down into carbon dioxide and water, releasing a large amount of energy. Ethylene can control changes during fruit ripening by regulating various physiological metabolic pathways of ripening at different levels, including metabolite synthesis, receptor sensing, signal transduction, and regulation of target gene expression by transcription factors. Therefore, scholars agree that ethylene is the most important endogenous regulator of the ripening process of climacteric fruits, including mango. In recent years, the effects of ethylene on the postharvest ripening of mango have been studied. Based on the sensitivity of mango to ethylene, these studies were mainly divided into two parts [10]. First, during the long-term storage and transportation of mangoes, the production of ethylene could be inhibited by physical and chemical methods so as to reduce the adverse effects of quick ripening and senescence. For example, 1-MCP and other ethylene inhibitors combined with low temperature could significantly inhibit the respiration rate of fruits and the decomposition of titratable acid while maintaining the hardness and quality of fruits, effectively extending the shelf life of fruits [11]. Secondly, in order to meet consumer demand, exogenous ethylene was used to stimulate fruits before marketing, drive the rapid synthesis of endogenous ethylene, and accelerate a series of physiological reactions of fruit ripening [12] and achieved a state of excellent quality of food and sales. For example, Ho et al. [12] placed ethylene α-cyclodextrin inclusion complex powder in the transportation process of mangoes, which could continuously release ethylene gas and stimulate the continuous production of endogenous ethylene in mangoes, thus shortening the postripening time of the fruit. In addition, the synthesis and metabolism of flavor, nutrients, and active substances in climacteric fruits are physiological processes triggered by the co-induction of intra-fruit respiration, ethylene release, and energy levels. Internal factors include the utilization and conversion of respiratory substrates, the transformation of respiratory pathways, the formation and conversion of energy intermediates, and the synthesis and regulation of secondary biomass [13].

### 2.2. Textural Changes in Mango during Ripening

Mango exhibits marked softening during postharvest ripening, which is mainly caused by the degradation of pectin and starch. On the one hand, pectin polysaccharides, cellulose, and hemicellulose in the cell wall undergo physiological changes, such as decomposition, de-esterification, and depolymerization, accompanied by marked decomposition and dissolution of neutral sugars, galacturonic acid, oligosaccharides, and residual sugar residues. Degradation of pectin components between connecting cells occurs, resulting in the enlargement of intercellular space, which in turn leads to the softening of the pulp tissue [14]. On the other hand, a large amount of starch distributed in the fruit cells is enzymatically hydrolyzed during ripening, thereby weakening the fruit cell support and loosening the cell structure.

The degradation of pectic components is associated with a series of changes in enzymatic activities. Prasanna et al. [15]. and Singh et al. [16] found that during the ripening of ‘Alphonso’ mangoes, free galacturonic acid content increased from 36 mg g^−1^ to 168 mg g^−1^ (fresh weight). Simultaneously, several pectin-degrading and pectin-modifying enzymes, including exogalacturonase, pectate lyase, pectin methylesterase, β-galactosidase, and α-arabinofuranosidase, were activated. These acted synergistically within the cell wall to reduce the molecular size of mango pectin polymers, leading to depolymerization and increased solubility of pectin polysaccharides. In addition, pectinesterases catalyze the de-esterification of methyl groups in acidic pectin in mature mangoes. Prasanna et al. [17] purified three β-galactosidase isomers from “Dashehari” mangoes and exogenously applied them on mangoes; they found that β-galactosidase degraded the endogenous pectins and softened the mango tissues. Furthermore, they identified *MiExpA1* and *MiPel1,* genes related to expansin and pectate lyase in mango, respectively, and found that the expression of these genes was highly correlated with the postharvest ripening characteristics of mango.

### 2.3. Color Changes and Pigment Synthesis in Mango

Color is the core indicator of fruit appearance quality and can be used to estimate the maturity of mangoes. The accumulation of pigments is the material basis for the formation of fruit color, and the type and content of pigments determine the color quality and color depth of fruits [18]. It has been reported that, owing to the synthesis and accumulation of anthocyanins and carotenoids, the epidermis of mature mangoes shows olive, orange, yellowish, dark red, purplish red, and yellow-red hues in different varieties [19]. Liang et al. [20] analyzed the carotenoid composition of the peel and flesh of “Tainong” and “Jin Huang” mangoes and found that the main carotenoids in the peel and flesh of mangoes included α-carotene and β-carotene, and to a lesser extent, lycopene, β-cryptoxanthin, zeaxanthin, and lutein.

Carotenoids are the most important pigments in mangoes, and numerous studies have elaborated on the mechanism and pathways of carotenoid accumulation during mango ripening [19]. Figure 3 illustrates the carotenoid synthesis pathway in dark-yellow- or orange-fleshed mangoes during postharvest ripening, which involves three main processes. The first process involves the synthesis of the carotenoid prerequisite, geranylgeranyl pyrophosphate (GGPP), by the mevalonate pathway (MEP). The second process involves the synthesis of all-trans carotenoids, that is, two GGPPs in mango form the first C40, a colorless phytoene, under the action of phytoene synthase (PSY), which is then successively dehydrogenated by phytoene dehydrogenase (PDS) to form δ-carotenoids. Next, all-trans lycopene is formed by δ- carotene desaturase (ZDS). Mahto et al. [21] noted that enzymes such as PDS, ZDS, and PSY and their related genes in the phytoene process are key factors in the expression of pigments during mango ripening. The third process is carotenoid epoxidation, which determines appearance and color and serves as a branching point for further anabolism of mango carotenoids. For example, the dark yellow or orange flesh of Tainong and Guifei mangoes is caused by the synthesis of intermediate phytoene through β-epoxidation and the accumulation of β-carotene. Pale-yellow-fleshed mangoes undergo α-epoxidation at the third step, resulting in the synthesis of α-carotenoids with lutein functional groups [22]. Currently, the pathways associated with pigment synthesis in mangoes with different skin and pulp colors have been extensively studied, but the establishment of a gene pool with the ability to regulate coloration is still lacking.

### 2.4. Flavor Changes during Ripening of Mangoes

#### 2.4.1. Degradation of Organic Acids during Postharvest Ripening of Mangoes

Organic acids are important components contributing to mango flavor. Gao et al. [23] analyzed the sugar and acid components of ten types of Yunnan mango varieties and reported that the organic acids of ripe mangoes mainly consist of citric acid, succinic acid, malic acid, and tartaric acid. Ma et al. [24] analyzed the sugar and acid components of eight varieties of mangoes and concluded that the organic acid composition of mangoes grown in different regions and of different varieties varied greatly, with purple-flowered mangoes and Fengshun seedless mangoes containing a considerably higher level of malic acid than citric acid and tartaric acid levels and the ‘*VanDyke*’ variety containing a higher level of tartaric acid. Usually, the organic acid content of postharvest mangoes is degraded substantially during the postharvest ripening process along with the climacteric change [25], which leads to the gradual loss of the edibility and flavor of the fruit. Organic acids are also involved in the tricarboxylic acid cycle of postharvest fruits and vegetables, yielding nicotinamide adenine dinucleotide phosphate (NADPH), and are therefore closely related to energy synthesis during postharvest ripening and senescence [26]. Organic acids are usually degraded more rapidly as preferential substrates for energy supply in postharvest fruit. Therefore, inhibiting the rapid degradation of organic acids during fruit storage and balancing the energy supply can effectively regulate the postharvest ripening process and maintain the poststorage quality of fruits [27]. However, there is a lack of detailed research and analysis on the synthesis and degradation patterns of organic acids in mangoes during postharvest ripening, and research in this area can contribute information on the energy supply of postharvest mango.

#### 2.4.2. Sugar Metabolism during Postharvest Ripening of Mangoes

Accumulation of soluble sugars provides sweetness to the fruit and enhances its commercial value. Soluble sugars in ripe fruit include other forms of sugars, such as xylose and maltose, in addition to glucose, fructose, and sucrose [28]. The changes in soluble sugars during the postharvest ripening of fruits are mainly attributed to sugar metabolism, which includes the hydrolysis of starch, accumulation of nonreducing sugars, and breakdown of nonreducing sugars to reducing sugars [29].

Metabolism of sugars in mango involves changes in the activities of several key rate-limiting enzymes. Castrillo et al. [30] showed that the activity of sucrose phosphate synthase increased approximately ten-fold during the rapid accumulation of sucrose in mangoes, and acid invertase activity increased rapidly at the beginning of ripening and decreased substantially during the later stage of postharvest ripening. Villar-Palasí et al. [31] found that the activity of fructose-1,6-bisphosphatase increased with the complete ripening of the fruit. In addition, the activity of glycolytic enzymes, namely hexokinase, reached a peak at the climacteric stage. Li et al. [32] investigated the mechanism of rapid sugar accumulation in “Tainong 1” and “Renong 1” mangoes and identified ten sugar metabolites, of which D-glucose was the most abundant. This was because the expression of two mango α-amylases increased significantly after ripening, and the expression of sucrose to D-glucose was facilitated by the upregulation of invertase and β-glucosidase. In addition, by inhibiting the expression of both *MYB* and *NAC* in the *TF* genome, the conversion of sucrose and D-glucose to other intermediate compounds was inhibited, which resulted in the retention of higher sugar content and sweetness in “Tainong 1” and “Renong 1” mangoes. Glucose metabolism is the process of supplying energy to the fruit, and there is a lack of analysis on the roles of sugar accumulation and breakdown during postharvest ripening and senescence in mangoes. An in-depth study of the main forms of sugar accumulation and enzymes related to sugar metabolism in mangoes can provide theoretical support for the development of new regulatory technologies for mango ripening.

#### 2.4.3. Synthesis of Aromatic Compounds during Postharvest Ripening of Mangoes

Mango is a typical aromatic fruit, containing more than 270 aromatic compounds, such as terpenoids, esters, and ketones. The main contributors are α-pinene, terpinene, 3-carene, cis-β-ocimene, limonene, benzaldehyde, dimethylstyryl, ethyl acetate, α-humulene, β-serine, acetophenone, and other compounds [33]. Fruit aromatic compounds are synthesized mainly with amino acids, fats, and carbohydrates as precursors through the metabolism of amino acids, fatty acids, carotenoids, and isoprenoids under enzyme catalysis [34]. Figure 4 illustrates the metabolic pathways of the two main types of volatile substances involved in mango, where ester aromatic compounds are generated by the β-oxidation of fatty acids, which provide a sweet aroma, and terpene aromatic compounds are converted from carbohydrates or fats through the isoprenoid pathway [35], which provide the characteristic aroma of mango.

Mango terpenoids consist mainly of hydrocarbons and their oxygen-containing derivatives with the isoprene structure (C_5_H_8_)_n_ as the basic unit. All terpenoids are formed by the condensation of the C_5_ structural unit isopentenyl pyrophosphate and its allyl isomers. Figure 4 highlights two key fundamental processes in the synthesis of fruit terpenoids. The first is the conversion of the two main synthetic precursors of terpenoids (including IPP and DMAPP) into key intermediate substrates, and the second is the reaction of key intermediate substrates to form volatile terpenes with characteristic aromas, such as monoterpenes, sesquiterpenes, triterpenes, squalenes, and sterols. Dudareva et al. [36] showed that although multiple branching occurs in terpenoid synthesis and different end products are produced, several branched pathways do not exist independently. Intermediate products are exchanged and supplemented during terpene biosynthesis, resulting in cross-linking that ensures the stability of the characteristic aroma in mango. In addition, although mango contains a sweet fruity aroma and fragrant volatile substances, there are few studies on the synthesis and regulation of aromatic compounds produced by fatty acid metabolic pathways and amino acid metabolism before and after mango ripening. An in-depth investigation of these areas can provide support for the construction of characteristic mango aroma profiles and the development of aroma recovery techniques after refrigerated storage and transportation.

### 2.5. Microbial Infection during Ripening of Mangoes

Generally, the growth and reproduction of infected microorganisms are not endogenous changes in the physiological metabolism of postharvest fruits. However, mango, banana, cherimoya, and other tropical fruits are more special. Their postharvest ripening process was usually bound to accompany the changes in anthracnose. There were two main pathogens of mango anthracnose reported. One is *Colletotrichum gloeosporioides* (Penz.) Sace; the other is *Colletotrichum acutatum* J. H. Simmonds [37]. The effects of mango and anthracnose on postharvest ripening were mutual. The large propagation of anthracnose would accelerate the postripening of mango, and the postripening of mango would also provide advantages for the onset of anthracnose [38]. Anthracnose had an obvious latent infection phenomenon, and it was basically impossible to determine whether the fruit was infected from the appearance of postharvested mangoes. It usually developed in the late ripening stage during the storage and transportation stage, leading to the rapid ripening and senescence of mango and eventually causing large-scale rotten fruits [39]. When mangoes passed the respiratory peak, anthracnose was in the early stage and the mango epidermis showed some black-brown spots and then rapidly expanded to round or irregular shapes of major disease spots. As ripening continued, the disease became severe, and the disease spots merged with each other. However, there was no clear causal relationship between postharvest mango ripening and anthracnose occurrence. In addition, a large number of studies on mango indicated that different preservation methods could inhibit mango anthracnose while inhibiting mango ripening [40,41]. Prusky et al. [42] used hot water brushing to treat postharvested mangoes and found that this brushing method could effectively reduce mango decay by *A. alternata* and prolong the storage period. Another research [43] of a combination of plant-defense-inducing chemical, inorganic salt, and hot water treatments on mango storage considered that integrating treatment could significantly inhibit anthracnose and delay mango ripening.

## 3. Methods and Techniques for Regulating the Preservation of Mangoes during Postharvest Ripening

### 3.1. Physical Techniques and Principles for Regulating Postharvest Ripening of Mangoes

Physical factors, such as temperature, humidity, air, and light, are the key environmental factors affecting the quality of fruits and vegetables during storage and transportation. Physical technology is currently the most researched and industrially applied postharvest preservation technology for fruits and vegetables, with the advantages of high safety, advanced equipment, and clear mechanisms [44].

#### 3.1.1. Low-Temperature Storage Techniques

Low-temperature storage is currently the most researched and widely used method for storing fruits and vegetables [28]. Several studies have shown that low temperatures can inhibit processes related to postharvest ripening, such as the metabolism of fatty acids, amino acids, sugar, and carotenoids, by regulating respiration and ethylene action [45]. Therefore, low-temperature storage can effectively prolong the storage period of fruits. In industrial applications, low-temperature storage at 13 °C is usually used to extend the freshness period of mangoes to approximately 20 d for improved poststorage quality. Li et al. [46] indicated that the use of low temperature combined with oxalic acid treatment could extend the storage period of mangoes to 49 d. In a study on “Hong Mang 6” mangoes, Gong et al. [47] found that low-temperature storage at 13 ± 1 °C significantly reduced the yellowing index, softening rate, weight loss, and decay rate of the fruit. It also delayed the synthesis of epidermal pigments and inhibited the softening process of fruit after ripening. Chen et al. [48] found that storage at 10 °C effectively inhibited fruit respiration and ethylene production, and this in turn reduced organic matter consumption and endogenous ethylene release, thereby extending the shelf life of the fruits. In addition, low-temperature treatment can also maintain the integrity of the mango cell wall and fruit firmness by directly reducing the activities of pectin methylesterase and polygalacturonase as well as delaying fruit softening and postharvest ripening [49,50]. However, it has also been suggested that cold-sensitive fruits, such as mangoes, lose their ability to ripen after prolonged cold storage, which may be due to cold damage that inhibits ripening-related enzyme activity and gene expression [51]. Nonetheless, Rosalie et al. [52] found that mango resistance increased under low-temperature storage at 7 °C, the water quality of active oxygen metabolism increased, and the carotenoid content of mangoes increased geometrically in the absence of low-temperature conditions, resulting in a significant increase in the physiological quality of mangoes. Some studies [50] have shown that the use of prestorage temperature stimulation and temperature acclimation treatments for tropical fruits, such as mango, banana, and papaya, can effectively induce and enhance fruit resistance, thus preventing the inability of fruit to ripen after storage. In the process of low-temperature storage, besides temperature, humidity, and ventilation, the stacking method might also affect the ripening of mango. Lehner and Siegmund [53] found that a dense arrangement of the fruits negatively impacted the ripening of mango and that better ventilation was a prerequisite for ripening and good flavor development for mango.

#### 3.1.2. Controlled Atmosphere Storage Technology

Controlled atmosphere storage (CA), under the premise of low-temperature storage, is a storage method that can reduce the physiological metabolism of fruits and slow down the senescence process of fruit ripening by regulating the concentration of O_2_, CO_2_, and other gases in the storage environment, thereby extending shelf life [54]. CA can effectively extend the storage period of tropical cold-sensitive fruits and maintain their postharvest ripening quality, and it has been widely used for the preservation of fruits and vegetables in China and abroad [54]. Wu et al. [55] and Wang et al. [56] found that high CO_2_ or low O_2_ concentrations reduced the sensitivity of mangoes to ethylene, thereby inhibiting the decomposition of organic matter, reducing the production of soluble sugars, and delaying the ripening process. Teixeira et al. [57] found that 1% O_2_ treatment could inhibit the activities of phosphofructokinase and pyruvate kinase and reduce the conversion of fructose-6-phosphate and phosphoenolpyruvate to fructose-1,6,diphosphate and pyruvate, thereby reducing the rate of respiration in mangoes. In addition, Mditshwa et al. [58] showed that hypoxia was able to block the linkage of ethylene to the receptors that trigger ripening, thereby inhibiting ethylene biosynthesis and delaying fruit after-ripening senescence.

#### 3.1.3. Low-Pressure Storage Techniques

Storage under low pressure can effectively reduce the respiratory intensity of fruits and vegetables, inhibit ethylene biosynthesis, delay the decomposition of chlorophyll, and inhibit nutrient consumption, thus delaying the ripening and aging of fruits and vegetables [59]. Currently, studies on the effect of low-pressure storage on postharvest mango ripening are relatively rare. Weng [60] found that the low-pressure preservation method delayed the appearance of the ethylene peak, reduced ethylene release, reduced the level of membrane lipid peroxidation, and maintained the stability of the fruit cell membrane structure. In addition, the low-pressure environment increased the activity of reactive oxygen scavenging enzymes in mangoes, maintained a high reactive oxygen scavenging capacity, and reduced the cellular damage caused by the accumulation of free radicals, thus inhibiting the postharvest ripening process of mango fruits and prolonging the storage period. Xie et al. [61] indicated that the main reason low-pressure storage can extend the shelf life of fruits and vegetables is that low pressure causes a hypoxic environment, regulating the release of ethylene from fruits. Moreover, reduced pressure can promote the outward diffusion of harmful volatile substances (such as hydrogen sulfide, thioether, and ammonia) from mango tissues, which reduces cell damage.

#### 3.1.4. Modified Atmosphere Packaging Technology

Modified atmosphere packaging (MAP) involves the modified composition of the internal atmosphere of a package, and this can delay the aging process of tropical fruits, such as mango. Ramayya et al. [62]. found that MAP with 50% CO_2_ significantly delayed the onset of respiratory peaks, slowed metabolism, and reduced oxidative decomposition of organic matter and nutrient consumption in mangoes, thereby delaying ripening and aging. Teixeira et al. [57] found that MAP inhibited the activity of aminocyclopropane carboxylic acid (ACC) synthase and ACC oxidase in mango, which reduced ethylene production, weakened the ability of ethylene to stimulate physiological effects, and considerably delayed fruit ripening and senescence. In addition, low O_2_ and high CO_2_ conditions inhibited chlorophyll degradation and delayed fruit color change [63]. MAP also reduced pectinesterase demethylation and polygalacturonase hydrolysis, thereby inhibiting the increase in soluble pectin (CSP) and the reduction in protopectin content and delaying the decline in fruit hardness and postharvest ripening senescence [64]. MAP is usually applied by packing the fruit in hermetically sealed polyethylene bags. However, polyethylene is nonbiodegradable and slow to decompose, which raises health and environmental concerns about solid waste after using this polymer [65]. As more and more countries advocated the use of biodegradable environmental protection materials in fruit and vegetable preservation, some MAP research began to use some biodegradable materials to manufacture MAP bags. Vilvert et al. [66] created a biodegradable bag with chitosan and graphene oxide and used it for mango storage. They found that the chitosan- and graphene oxide-based biodegradable bags were effective in delaying ripening, especially in reducing weight loss, respiration rate, and anthracnose incidence. Therefore, further research on mango thermal stimulation can provide strong technical support for the long-term low-temperature storage and export of mango.

#### 3.1.5. Heat Treatment Techniques

Heat treatment, also known as thermal stimulation treatment, is a common pretreatment method for the commercialization of fruits and vegetables, and the technology is well-established for the storage and preservation of a wide range of fruits and vegetables [67]. The main methods of heat treatment for fruits and vegetables in the industry include hot air, hot steam, hot water, hot ash burial, far infrared, and microwave treatment [68]. Heat treatment time is an important factor affecting the effect of this technology. When the heat treatment time changes, the heat shock reaction effect of fruits is also different [69]. The stress resistance of mango cells was enhanced under short-term heat shock, and the self-protection mechanism of mango fruit was improved, which could delay ripening and improve storage quality. Seid et al. [70] used hot water to soak mangoes for a short time. It was found that the stimulation of hot water could effectively inhibit the transpiration and respiration rate of mangoes and then slow down the degradation of soluble solids and organic acids, delay fruit ripening, and prolong storage time. Besides inhibiting respiration, hot water treatment could also significantly inhibit the degradation of the fruit cell wall and delay ripening. Li et al. [71] treated papaya with hot water at 54 °C, which also significantly inhibited the respiration rate of fruits and effectively inhibited the degradation of primary pectin to soluble pectin, thus maintaining the hardness of mango and delaying the ripening. In addition, Lin et al. [72] treated Jinhuang, Aiwen, Ivory, and Tainong No. 1 mangoes with hot steam and found that hot steam could delay the ripening of the four kinds of mangoes and maintain the storage quality of exported mangoes. As a traditional preservation treatment, thermal stimulation has many different treatment methods. It can not only regulate the temperature and processing time but also combine other treatment methods to delay the ripening of fruits and vegetables [73]. The results of Sripong et al. [73] suggested that the combination of hot water and UV-C treatment may be used as a method not only for suppressing anthracnose disease but also for delaying the ripening of harvested mangoes by inducing the expression of plant-defense-related genes. In addition to hot water and hot steam, hot air stimulation also has a positive stimulating effect on the physiological metabolism of fruits and vegetables [74]. However, there is a lack of relevant research on hot air treatment of mango.

However, it has also been reported that high-temperature treatment of fruits may have negative effects on postharvest storage quality. For instance, Wang et al. [75] found that hot air treatment at 38 °C for 24 h and 48 h enhanced membrane lipid peroxidation, which severely damaged the integrity of mango cell membranes and accelerated fruit senescence. Currently, heat treatment is mainly an important method of alleviating chilling injury for cold-sensitive fruits during low-temperature storage. However, there is little research on the effect of heat treatment on the ripening of mangoes during low-temperature storage. Through metabolomics analysis of mango peel, Vega-Alvarez et al. [76] found that hot water treatment could effectively maintain a higher level of galloylquinic acids, gallic acid esters, gallotannins, quercetin 3-O-rhamnoside, mangiferin, myo-inositol, linolenic acid, and sugar content. They deduced that hot water treatment could improve the cold tolerance of mango and maintain normal ripening after low-temperature storage, possibly because hot water treatment improved the antioxidant activity of mango. In-depth research in relevant areas can provide theoretical support for the development of new storage and transportation technologies for the mango import and export industry.

#### 3.1.6. Irradiation Treatment Techniques

Irradiation technology is a common nonthermal processing technology of food. It is a process of energy transfer. According to the different effects of processing objects, it can be divided into nonionizing irradiation and ionizing irradiation. Nonionizing irradiation mainly uses the thermal effect produced by rays (including radio frequency, microwave, infrared light, visible light, etc.). Ionization irradiation mainly uses the ionization effect produced by rays (including ultraviolet, X-ray, γ ray, particle ray, etc.) [77]. At present, the main irradiation technologies commonly used in postharvest fruits and vegetables are electron beam irradiation, ultraviolet irradiation, and visible lighting.

Ultraviolet irradiation (UV) is the most common fruit and vegetable preservation irradiation technology, which has technical universality. It has been applied to postharvested mango, citrus, apple, strawberry, and other fruits [78]. Both UV-C and UV-B irradiation have been reported to have effects on mango preservation and the delay of ripening through direct and indirect factors. Ruan et al. [79] pointed out that UV-B treatment could induce cold resistance and delay the ripening process by stimulating the antioxidant system of mangoes. Sripong et al. [73] have pointed out that UV-C treatment significantly delayed fruit ripening by maintaining fruit firmness, retarding the progressive increase in total soluble solids and delaying the decrease in titratable acidity. González-Aguilar et al. [80] pointed out that UV-C treatment could effectively increase the activity of phenylalnine ammonialyase and lipoxygenase, thus keeping a lower maturity and decay rate after storage for 18 d. On the other hand, it has been reported that UV-C treatment can significantly inhibit conidial germination and sporulation of colletotrichum and delay the ripening process of mango [81]. Although studies on UV irradiation of mango were abundant and effective, an investigation of the main mango-producing areas in China showed that UV irradiation was still very rare in the practical production and application of mango storage and preservation. How to improve the application feasibility of UV irradiation is worthy of research.

Electron beam radiation (E-Beam) is a new irradiation technology, which uses the electromagnetic field of the accelerator to produce a relatively high-energy electron beam for food processing. At present, E-Beam has been more widely used as an insecticidal, bacteriostatic, and germinating treatment technology for agricultural products [82]. But as a fruit and vegetable preservation technology, the research and application on mango is not rich. Studies indicated that E-Beam could inhibit the ripening of postharvested fruits and vegetables mainly by inhibiting respiration and enhancing antioxidant capacity [82]. Nguyen et al. [83] treated mangoes with 0.5 kGy E-Beam and found that the respiration rate and ethylene production of the fruit increased immediately triggered by E-beam treatment. However, the respiration rate was significantly suppressed compared to those fruits with nontreatment until the end of storage, and the E-Beam-treated mangoes showed a longer storage period. Dong et al. [84] pointed out that E-beam treatment with a dose of more than 1.0 kGy could not delay the ripening process of mango but accelerated the browning and decay of mango. Therefore, it is necessary to pay attention to the dosage when using E-beam treatment as a preservation method to delay the ripening of different varieties of mango.

As nonionizing irradiation, visible lighting acted on fruit and vegetable preservation technology for a shorter time than the other two irradiation technologies, and the mechanism was still not clear. In recent years, some studies [85] have found that light is also one of the environmental factors that can affect the senescence of fruit after ripening. It has been reported that light can regulate ripening by regulating skin color change and pigment synthesis [85]. Red light irradiation at 500 Lx stimulated β-carotene accumulation in “Small Tai Mang” mangoes, caused earlier respiratory peaks, and promoted soluble sugar accumulation, thereby increasing the rate of hardness reduction by 33.39% after storage compared with that observed in the control group. The results of the study indicate that red light stimulation could activate intracellular pigment accumulation in mangoes and accelerate the postharvest ripening process. Studies on the application of light treatment on fruits, such as oranges [86] and peaches [87], have shown that light can promote or inhibit the ripening of fruits and vegetables by regulating key genes associated with the syntheses of carotenoid and anthocyanin. However, the key regulatory pigments and limiting enzymes or genes associated with the protective effects of light on mango have not been identified.

### 3.2. Chemical Preservatives and Principles of Regulating the Postharvest Ripening of Mango

Chemical preservatives have become a popular quality control technology for postharvest preservation of fruits and vegetables due to their low cost and ease of use. Currently, in China, the national standards do not specify the category of chemical preservatives but only classify them as food additives in the field [2].

#### 3.2.1. Calcium Salt Treatment Technology

Ca^2+^ is both a secondary messenger in the signal transduction pathway of the plant and a regulator of plant hormones as well as biotic and abiotic stresses [88]. It acts by maintaining fruit firmness, reducing fruit respiratory intensity, decreasing ethylene release, and inhibiting the postharvest ripening of fruits [88]. The mechanism of action of Ca^2+^ on fruits and vegetables has been relatively well studied, mainly including the following three aspects. First, Ca^2+^ can bind to the RCOO- group of polygalacturonic acid in fruit and vegetable cells, forming an interchangeable form and regulating membrane permeability and related processes. Second, Ca^2+^ binds to organic macromolecules to form bound calcium, increasing the physical rigidity and stability of the cell wall structure. In addition, Ca^2+^ in cells also regulates the dynamic balance of intracellular and extracellular ions and the structure and function of the plasma membrane, ensuring normal physiological metabolisms of cells [88]. Ngamchuachit et al. [89] found that Ca^2+^ treatment delayed the degradation of mango cell wall pectin and maintained the stability of phospholipid and protein binding, thus delaying the decline in fruit hardness. Ca^2+^ treatment also inhibited the activity of mango polyphenol oxidase, thus inhibiting the browning of mango fruit flesh and delaying fruit senescence. Jiang et al. [90] found that 1% calcium treatment had an inhibitory effect on the postharvest ripening of mango. Common calcium salts used for postharvest treatments include calcium chloride, calcium lactate, calcium nitrate, calcium ascorbate, calcium amino acids, and calcium sugar alcohol [67]. Of these, calcium chloride treatment is the most effective in delaying the storage cycle, postponing fruit softening, and improving freshness quality [91].

#### 3.2.2. Oxalic Acid and Oxalate Treatment Technologies

Oxalic acid and oxalates are common organic acids found in plants. In recent years, oxalic acid has attracted great attention in the postharvest storage of mangoes and is widely believed to delay postharvest senescence and extend the storage period of mangoes [92]. Huang et al. [93] found that 1% chitosan and 5 mmol/L oxalic acid and oxalic acid–chitosan hybrid treatments maintained the sensory and nutritional quality of postharvest mango fruits, improved antioxidant capacity, and delayed the process of postharvest senescence. Moreover, they reported that oxalic acid–chitosan hybrid treatment significantly inhibited the accumulation of anthocyanins in mango epidermis and induced the accumulation of flavonoids in fruit pulp. Zheng et al. [94] showed that the treatment of mango with oxalic acid reduced the content of membrane lipid peroxidation products and inhibited cell wall hydrolase activity and expansin expression, thereby maintaining the stability of membrane structure and delaying the fruit ripening process. Zhang [95] studied “Hong Mang 6” mango and found that oxalic acid treatment could improve the ability of the ASA-GSH cycle to scavenge free radicals and reduce the oxidative damage of cell membranes, proteins, nucleic acids, and other macromolecules after harvesting, which delayed the ripening process.

In addition, several studies have reported that erythorbic acid [96], plant essential oils [97], and plant water-soluble extracts [98] improve the storage quality of mangoes, but no study has clearly identified the significant effects on the specific physiological metabolism associated with the postharvest ripening of mango. Therefore, this is the major factor limiting the application of these natural or synthetic chemical preservatives in the mango industry.

#### 3.2.3. Edible Coating Technology

In recent years, research on edible coating films for fruit and vegetable preservation has increased as high-safety and green chemical preservation techniques are explored [99]. Currently, commonly used coating materials for mango preservation are chitosan, konjac glucomannan (KGM), sodium alginate (SA), and carboxymethyl cellulose (CMC), as well as composite coatings of these materials [100]. These nontoxic coating films can control gas entry, reduce water transpiration, and inhibit respiration by isolating mangoes from the external environment. Ngo et al. [101] used a pectin-nanochitosan coating to preserve the freshness of “Elephant” mangoes and found that 2% pectin-nanochitosan effectively reduced water loss from the mangoes and delayed the browning and hardness loss of the fruit peel. The application of CMC coating to “*Ratol-12*” mangoes at room temperature not only reduced the increase in the activities of pericarp cellulase, pectin methylesterase, and polygalacturonase in the peel but also significantly increased the activity of peroxidase, catalase, ascorbate peroxidase, and superoxide dismutase in the fruit, thus enhancing the level of reactive oxygen metabolism and delaying postharvest ripening [102]. In addition, with the continuous development of functional food ingredients, some new edible coating techniques have emerged. Xu and Wu [103] found that the fucoidan coating film can delay the peak of mango respiration, reduce water loss and consumption of soluble solids, inhibit postharvest ripening, and prolong its freshness. Ma [104] treated “Tainung” mangoes with shellac compound tannins and found that the compound coating had a higher coverage density on the mango peel than the traditional coating, which slowed down the respiration rate and transpiration of the fruit, significantly inhibited Polyphenol oxidase (PPO) activity of the mango, and delayed epidermal browning. In addition, some studies indicated that composite coatings might have better bacteriostatic properties and preservation effects than single coatings. At the same time, the composite coating might also have good thermal stability and can protect the structure of the fruit to a certain extent. Zhou et al. [97] used a certain proportion of CMC/PL-8% GEO composite membrane to treat mangoes, which could extend the shelf life of mangoes to 9 days at room temperature and maintain low maturity. The addition of natural active substances such as essential oils could play a role in the preservation of fruits during storage. Dong et al. [105] and Raina et al. [106] demonstrated that GEO not only had bacteriostasis but also could improve the antioxidant effect of fruits and effectively delay the ripening of mango. Xiao et al. [107] found that a new chitosan/zein-cinnamaldehyde nanocomposite film could not only reduce the respiration rate but also the retardation of yellowing obviously. He pointed out that composite coatings could also increase the adhesion of essential oils and other substances. Currently, the postharvest application of traditional edible coating technology on mango has been widely adopted, and some new materials have been exploited. Therefore, edible coating films have a broad development prospect in the postharvest preservation of mangoes. 

### 3.3. Role and Mechanism of Plant Growth Substances in Regulating Postharvest Ripening in Mango

Ethylene, auxin, abscisic acid, and jasmonic acid are important endogenous plant hormones that regulate the postharvest ripening of mango [108]. Different endogenous hormones have different effects on the regulation of postharvest ripening in mango.

#### 3.3.1. Ethylene and 1-Methylcyclopropene Regulation Techniques

As a typical climacteric fruit, numerous studies have indicated a series of postharvest physiological changes in mango accompanied by elevated ethylene release. Ethylene regulates various ripening phenomena in fruits by regulating the expression of genes with different processes, such as chlorophyll degradation, carotenoid synthesis, starch conversion to sugar, and cell wall regulation [109]. It is a versatile phytohormone that can promote ripening and senescence in postharvest fruits and vegetables. Trace amounts of ethylene can activate enzymes related to pectin degradation and sugar metabolism to rapidly promote ripening; in mangoes, 0.01 ppm ethylene can trigger postharvest ripening [110]. The release of ethylene from unripe mangoes increases rapidly, and different varieties of mangoes show different patterns of respiration and ethylene release [111]. The process of mango ethylene synthesis mainly involves ACC synthase (ACS) and ACC oxidase (ACO). ACS catalyzes the conversion of S-adenosylmethionine (SAM) to ACC, which is then oxidized to ethylene by ACO. Wu [112] indicated that the expression of multiple ACC synthase genes was induced during the early stage of fruit ripening, leading to increased ACC production. In most cases, the activity of ACS determines the rate of ethylene biosynthesis.

Numerous studies have shown that the control of ethylene-related enzymes and their gene expression can regulate the postharvest ripening of mango and achieve extended freshness [111]. Traditional methods of ethylene control, such as the use of exhaust gas, potassium permanganate oxidation, carbon bromide adsorption, catalytic oxidants, and low-pressure storage, need to be supplemented with appropriate refrigeration conditions, and these control methods often have high economic costs or poor practicality, resulting in low usage in actual production [109]. In addition, the use of photocatalytic titanium dioxide for the complete oxidation of ethylene to achieve ethylene control has been proposed as a low-cost and efficient method, which does not only allow for the precise control of ethylene content but also has the advantage of low energy consumption [113].

During fruit ripening, exogenous application of ethephon increases the respiration rate, ethylene production, and fatty acid content of fruits, whereas 1-methylcyclopropene (1-MCP), as an ethylene receptor competitive inhibitor, can effectively reduce ethylene synthesis [114]. 1-MCP, a gaseous compound, is widely used in the preservation of mangoes, apples, bananas, persimmons, and other fruits and vegetables due to its high efficiency, activity, and safety [11]. Several studies have shown that 1-MCP can inhibit ethylene biosynthesis and respiration, maintain fruit firmness, inhibit the increase in soluble solids, and maintain the storage quality of fruits [34,107]. Other chemicals can also inhibit ethylene production and preserve the freshness of fruits. For example, Zaharah and Singh [115,116] applied NO exogenously to delay the climacteric rise in ethylene in mangoes and reduce the respiration rate, confirming the important role of ethylene in the ripening of fruits.

#### 3.3.2. Indole-3-Acetic Acid

Indole-3-acetic acid (IAA), also known as auxin, plays a crucial role in fruit development [117]. IAA can have a dual effect during fruit ripening; it can promote ethylene production and release while delaying fruit ripening. The stimulation of ethylene synthesis is due to the induction of ACS by IAA [118], which indirectly affects the ripening process. The delay in ripening may be attributed to the effect of IAA on the genes associated with ripening in mango. Zaharah et al. [119,120] showed that endogenous IAA levels in mango were higher before the climacteric period and rapidly decreased after this period. Compared with ethylene and abscisic acid treatment, IAA treatment did not promote fruit softening but instead kept the fruit firmer. Singh et al. [121] reported that IAA treatment had no effect on the gene expression of enzymes related to ripening in mango, such as alcohol dehydrogenase (ADH) and 4-hydroxyphenylpyruvate dioxygenase (HPPD). However, El-Sharkawy et al. [122] noted that IAA regulates the expression of many genes involved in biosynthesis, transport, and signaling in mango in addition to indirectly upregulating ethylene synthesis during the climacteric period. It has been hypothesized that IAA may be involved in the ripening process. Wang et al. [123] found that the signaling pathways of auxin and ethylene synthesis were regulated by IAA and ERF transcription factors in peaches. Two genes, *PpIAA1* and *PpERF4*, promoted peach ripening by targeting the gene promoters that bind to auxin and ethylene and enhancing their activities.

#### 3.3.3. Abscisic Acid 

Exogenous abscisic acid (ABA) can stimulate ethylene biosynthesis in climacteric fruits, such as mango, tomato, apple, and banana, and shorten the time required for fruit ripening [124]. Zaharah et al. [120] found that exogenous ABA activated ACS and ACO ethylene-metabolizing enzyme activities in mango, increased ethylene release and the peak respiratory rate, and promoted fruit color change and softening. Qi et al. [125] reported that the firmness and titratable acid fraction of ABA-treated fruits were significantly lower than those of the untreated group. The activities of cellulase and pectin methylesterase in the fruit were higher, and ABA stimulated the expression of genes, such as *ERS2* for ethylene signal transduction, *ETR2*, *ERF3*, *ERF4*, and *ERF5*. ABA, as an endogenous phytohormone that regulates ripening, has been more extensively studied in the postharvest field of fruits and vegetables. However, the specific regulatory mechanism of the metabolism of postharvest ripening in mango has not been elucidated. There is a need for further research to guide the elucidation of the pathways for other endogenous phytohormones regulating the postharvest senescence of fruits and vegetables.

#### 3.3.4. Jasmonic Acid and Methyl Jasmonate

Jasmonic acid and methyl jasmonate (MeJA) belong to a class of oxidized lipids in plant endogenous linolenic acid that plays a role in the development and ripening of many fruits [126]. Jasmonic acid usually interacts with other endogenous hormones to influence fruit respiration and ethylene release, thereby regulating fruit ripening [127]. Before fruit ripening, the level of endogenous jasmonic acid increases, and this promotes ethylene biosynthesis, the expression of signal transduction genes, and ethylene release. In contrast, after fruit ripening, high jasmonic acid content leads to a reduction in ethylene biosynthesis [126]. Gong et al. [128] treated green ripe mangoes with exogenous MeJA, which improved fruit color and flavor, induced resistance, and slowed the rate of decline in fruit firmness, thereby improving storage quality. However, current research on the postharvest effect of jasmonic acid has focused on disease resistance and response to environmental factors, and there is a relative lack of research on postharvest ripening regulatory pathways and genome mining [129].

#### 3.3.5. Clopyralid

Clopyralid (Forchlorfenuron, CPPU) is a plant growth regulator that functions as a cytokinin, competing with zeatin to bind to cytokinin oxidase, and it plays an important role in maintaining endogenous cytokinin levels [130]. Its physiological activity is higher than that of purine cytokinins, and it has biological effects, such as the promotion of plant cell division and enlargement and inhibition of senescence [131]. Zhang et al. [132] found that 10 mg/L CPPU treatment did not only significantly delay yellowing, decrease firmness, and lead to the decomposition of SSC and titratable acids of “Guifei” mangoes during storage at room temperature but also inhibit the respiration rate and ethylene production of the fruits. Moreover, CPPU treatment also inhibited β-Gal activity to a greater extent, thereby maintaining pectin stability and inhibiting mango softening. Further research revealed that the interaction between CPPU and ethylene in fruit is complex and may be influenced by various factors, such as fruit type and variety, CPPU dose, treatment period, treatment method, and storage conditions [133,134].

#### 3.3.6. Melatonin

Melatonin (MT) is one of the research hotspots in the postharvest field in recent years, and it has been verified to be a phytohormone-like signaling molecule that can participate in the regulation of various physiological processes in plants, including seed germination, flowering, circadian rhythms, photosynthetic productivity, ripening and senescence, and response to environmental stresses [135]. MT treatment can regulate mango ripening and softening by reducing ABA synthase activity and ABA accumulation [136]. Liu et al. [137] found that MT treatment effectively inhibited the activities of mango pectin-modifying enzyme, β-galactosidase, and pectin methyl esterase, limited the depolymerization of pectin polysaccharides, and regulated mango fruit ripening and softening by inhibiting ethylene and ABA biosynthesis. In addition, melatonin has strong antioxidant effects and may have an inhibitory effect on free radical production. Dong et al. [138] treated mangoes with 200 μM MT and found that MT delayed the degradation of phosphatidylglycerol and phosphatidylinositol, hindered the accumulation of phosphatidylserine and phosphatidic acid in membrane phospholipids, and reduced the levels of H_2_O_2_ and malondialdehyde in the fruit exocarp, thus protecting mango cell membrane integrity and delaying the ripening process.

### 3.4. Comparison of Different Physical, Chemical, and Biological Treatments on Postharvest Mango Ripening

As shown in Table 1, Table 2 and Table 3, a total of seventeen physical, chemical, and biological treatments have positive or negative effects on the ripening and senescence of postharvest mango. Most treatments mainly inhibited mango ripening, while there were few treatments that had a ripening effect on mango. Due to the higher safety of physical treatments, they are a more common and feasible way to affect the ripening of postharvest mango. There are many pieces of research on chemical preservative treatment in fruit and vegetable preservation, but there are few preservative treatments that could directly affect the ripening of postharvest mango. At present, edible coating is an increasingly studied method in the chemical treatment of mango ripening. However, the main reason for limiting the application of edible coating technology to mango and other fruits is the additive permission laws and regulations in various countries. Biological treatment is mainly regulated for mango ripening by some endogenous plant hormones. Ethylene and abscisic acid were the main biotreatments to promote mango ripening, while the other four hormones inhibited mango ripening. In addition, besides endogenous hormones, plant extracts and beneficial microorganisms are also the focus of current biological preservation technology research [6]. However, studies on the effects of microorganisms and plant extracts on the ripening and senescence of postharvest mango are still severely lacking. At present, the ripening regulating methods for postharvest mango storage, transport, and sale are limited, and most of them are still in the research stage [2]. Therefore, how to improve the applicability of these methods was also a worthy research direction.

## 4. Conclusions

With the global expansion of the mango industry and evolving consumer demand, the study of the postharvest ripening of mangoes is of high research value. The postharvest physiological changes in mango are a complex process, involving ethylene release, respiration, energy supply, sugar metabolism, acid degradation, terpene volatile synthesis, carotenoid accumulation, and degradation of cell wall pectin. Existing studies have made progress in epiphenomena, metabolites, and key rate-limiting enzymes, laying the foundation for the development of postharvest ripening regulation technologies for mango. However, further research is required on the physiological and metabolic mechanisms of postharvest ripening of mango at the cellular, protein, and gene levels using modern biotechnology. The regulatory technologies for the postharvest ripening of mango are mainly divided into three aspects: physical, chemical preservatives, and endogenous biological hormones. The physical postharvest ripening technologies of mango are relatively old, mainly involving the use of low temperature and air conditioning to inhibit mango postharvest ripening during storage. Studies on novel irradiation, light, temperature acclimation, and physical preprocessing techniques are relatively lacking. Chemical preservatives for the regulation of postharvest ripening in mango are limited; only 1-MCP and calcium salts are clearly able to regulate metabolic processes associated with the postharvest ripening of mango. Although some studies [125] have shown that oxalate and edible coatings can inhibit ripening, few studies [127] have elucidated the specific regulatory mechanisms. The use of phytohormones is the most important technique for regulating the postharvest ripening of mangoes, and the hormones, including ethylene, ABA, IAA, CPPU, and MLT, can promote or inhibit postharvest ripening in mangoes. With the development of the field of plant physiology and biochemistry, an increasing number of endogenous plant hormones are being developed and applied. Although there is abundant research on the mechanism and technology of postharvest ripening in mango, there is a lack of relevant research in gene bank establishment, gene editing, variety mapping, metabolism and physiology network, postharvest ripening prediction models, and novel storage and transportation technology. These areas need to be explored by scholars to provide theoretical and technical support for the development of the mango industry.

## Figures and Tables

**Figure 1 foods-12-00173-f001:**
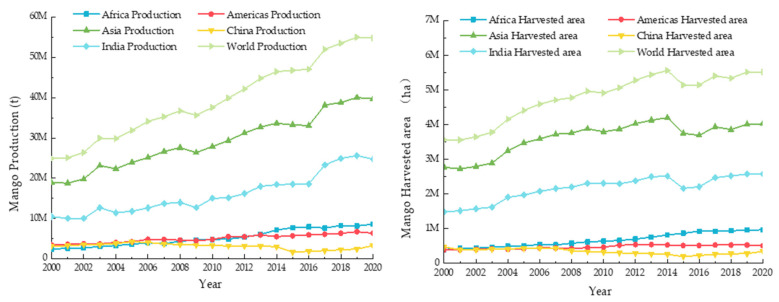
Mango planting area and yield in the world from 2000 to 2020 [3].

**Figure 2 foods-12-00173-f002:**
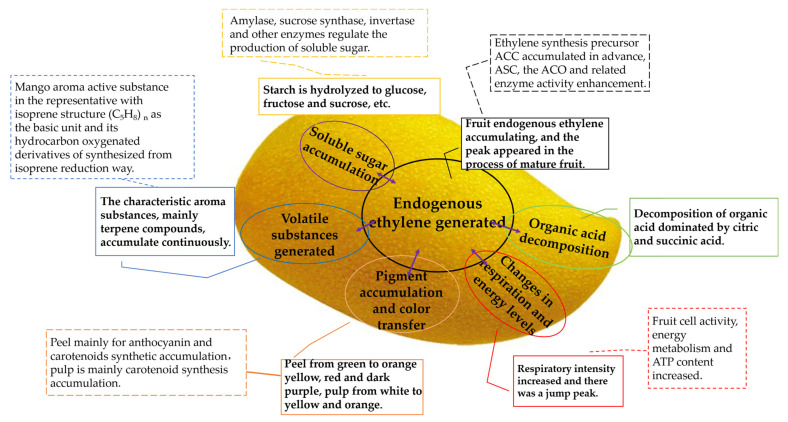
The main physiological changes in mangoes during ripening.

**Figure 3 foods-12-00173-f003:**
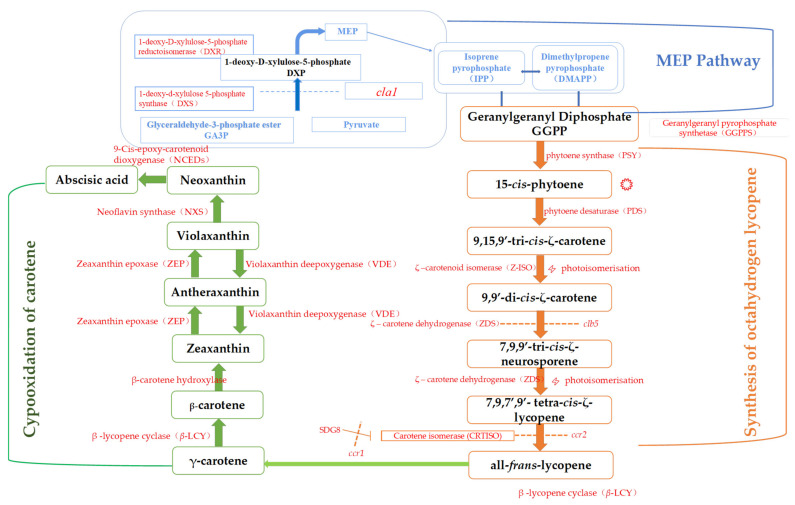
Carotenoid synthesis pathway in mango with red and orange color.

**Figure 4 foods-12-00173-f004:**
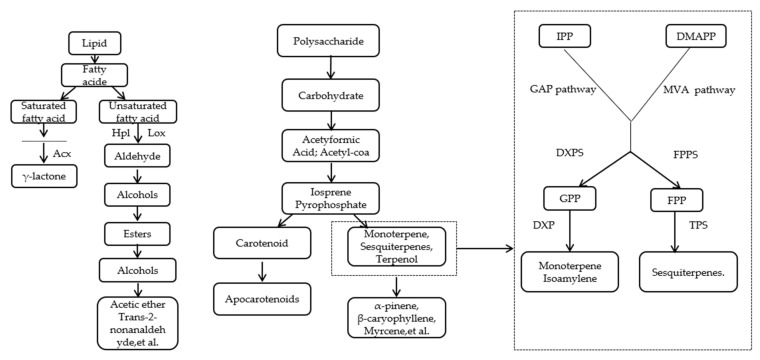
Metabolic pathways and main components of mango aroma substances.

**Table 1 foods-12-00173-t001:** Effects and comparison of different physical treatments on postharvest mango ripening.

Treatment	Effect on Ripening of Postharvest Mango	Other Description	Reference
Low-temperature storage	It could delay ripening, reduce the rate of respiration and ethylene release, prolong storage period, and maintain hardness and color quality after storage effectively.	It is the most common method used for mango preservation, storage, and transport. However, during long storage, low-temperature storage may cause chilling injury to mango.	[48,49,50,51,52]
Controlled atmosphere storage (CA)	It could delay ripening, reduce the rate of respiration and ethylene release, and extend storage period longer than low-temperature storage.	It is more suitable for the long storage of mangoes and other tropical fruits. The CA usually inhibited the chilling injury of mango and required precise and rigorous air conditioning equipment.	[55,56,57,58]
Low-pressure storage	It could delay ripening effectively, reduce the respiratory intensity and ethylene release, and maintain a high reactive oxygen scavenging capacity.	Low-pressure storage is usually achieved on the basis of low-temperature storage by combining vacuum and air conditioning. This technique has been less studied for mango storage.	[60,61]
Modified atmosphere packaging (MAP)	It could delay ripening. MAP wrap is usually gas-selective permeable, and it achieves a low oxygen and high carbon dioxide environment through mango respiration. Hence, it reduces the respiratory intensity effectively.	It is an environmentally friendly technology, but the research and application of MAP on mango storage were rare.	[57,62,66]
Heat treatment	It could improve mango self-protection mechanism, effectively inhibit the transpiration and respiration rate, reduce the cell wall degradation, and delay ripening. However, inappropriate heat shock treatment would enhance membrane lipid peroxidation and accelerate mango ripening and senescence.	It was also known as thermal stimulation treatment. The heat shock treatment usually includes hot steam, hot water, and hot air.	[70,72,73,75]
Ultraviolet irradiation (UV)	Both of the UV-B and UV-C treatment could delay the ripening of mango. The UV-B could also stimulate the antioxidant system and induce cold resistance. The UV-C treatment can significantly inhibit conidial germination and sporulation of colletotrichum and delay the ripening process of mango.	The UV treatment is one of the irradiation treatments. However, limited by UV preservation equipment, how to apply UV technology for mango ripening and preservation is an important problem.	[79,80,81]
Electron beam radiation (E-Beam)	It could delay ripening and prolong storage period. The respiration rate and ethylene production of the fruit increased immediately, triggered by E-beam treatment. However, the respiration rate was significantly suppressed until the end of storage. The E-beam treatment with a dose of more than 1.0 kGy could not delay the ripening process of mango but accelerated the browning and decay of mango.	As a fruit and vegetable preservation technology, the research and application on mango were not rich.	[82,83,84]
Visible lighting	The red light stimulation might activate intracellular pigment accumulation in mangoes and accelerate the postharvest ripening process.	It is a nonionizing irradiation treatment. Different wavelengths of visible light may produce different effects on fruit ripening.	[44,85]

**Table 2 foods-12-00173-t002:** Effects and comparison of different chemical treatments on postharvest mango ripening.

Treatment	Effect on Ripening of Postharvest Mango	Other Description	Reference
Calcium salt treatment	It mainly maintained fruit firmness by delaying the degradation of mango cell wall pectin and then inhibited postharvest ripening of mango.	Common calcium salts used for postharvest treatments include calcium chloride, calcium lactate, calcium nitrate, calcium ascorbate, calcium amino acids, and calcium sugar alcohol.	[88,89,90]
Oxalic acid and oxalate treatment	It could delay postharvest ripening and extend the storage period of mangoes. Oxalic acid reduced the content of membrane lipid peroxidation products and inhibited cell wall hydrolase activity and expansin expression, thereby maintaining the stability of membrane structure and delaying the fruit ripening process.	It is a kind of common organic acid preservative, which has abundant preservation research basis and good prospect in the application of mango preservation.	[92,93,94,95]
Edible coating	Coating films can control gas entry, reduce water transpiration, inhibit respiration by isolating mangoes from the external environment, and then delay the ripening.	Commonly used coating materials for mango preservation are chitosan, konjac glucomannan (KGM), sodium alginate (SA), and carboxymethyl cellulose (CMC), as well as composite coatings of these materials. However, the main reason for limiting the application of edible coating technology to mango and other fruits is the additive permission laws and regulations in various countries.	[97,101,102,103,104,105,106,107]
Ethylene treatment	Ethylene regulates various ripening phenomena in mangoes by regulating the expression of genes with different processes, such as chlorophyll degradation, carotenoid synthesis, starch conversion to sugar, and cell wall regulation.	Ethylene treatment is both a chemical and biological preservation method. It is the most common method to accelerate ripening at present, which is widely used in the storage, transportation, and sale of mango. Ethephon is a common ethylene preservative.	[109,110,111,112]
1-methylcyclopropene treatment (1-MCP)	1-MCP, as an ethylene receptor competitive inhibitor, can effectively reduce ethylene synthesis and then delay the ripening of mango.	1-MCP is a gaseous compound and widely used in mangoes preservation due to its high efficiency, activity, and safety.	[11,109,110,111,112,113,114]

**Table 3 foods-12-00173-t003:** Effects and comparison of different biological treatments on postharvest mango ripening.

Treatment	Effect on Ripening of Postharvest Mango	Other Description	Reference
Ethylene treatment	Ethylene regulates various ripening phenomena in mangoes by regulating the expression of genes with different processes, such as chlorophyll degradation, carotenoid synthesis, starch conversion to sugar, and cell wall regulation.	Ethylene treatment is both a chemical and biological preservation method. It is the most common method to accelerate ripening at present, which is widely used in the storage, transportation, and sale of mango. Ethephon is a common ethylene preservative.	[109,110,111,112]
Indole-3-acetic acid (IAA)	It is also known as auxin and plays a crucial role in fruit development and ripening. IAA can have a dual effect during fruit ripening. It can promote ethylene production and release while delaying fruit softening.	As a legitimate plant growth hormone, it is seldom used in the preservation of mango.	[119,120,121,122,123]
Exogenous abscisic acid (ABA)	ABA can stimulate ethylene biosynthesis in mango and shorten the ripening process.	ABA, as an endogenous phytohormone, is used in the postharvest field of fruits and vegetables.	[120,124,125]
Jasmonic acid and methyl jasmonate (MeJA)	Jasmonic acid usually interacts with other endogenous hormones to influence fruit respiration and ethylene release, thereby regulating fruit ripening. Before fruit ripening, jasmonic acid could promote ethylene biosynthesis and then promote ripening. In contrast, after fruit ripening, high jasmonic acid content leads to reduction in ethylene biosynthesis and delayed ripening.	Current research on the postharvest effect of jasmonic acid has focused on disease resistance and response to environmental factors, and there is a relative lack of research on postharvest ripening regulatory pathways and genome mining.	[126,127,128]
Clopyralid (Forchlorfenuron, CPPU)	It can effectively inhibit mango ripening, especially for mango softening.	CPPU is a plant growth regulator that functions as a cytokinin.	[132,133,134]
Melatonin (MT)	MT treatment can delay mango ripening and softening by reducing ABA synthase activity and ABA accumulation. In addition, melatonin has strong antioxidant effects and may have inhibitory effect on free radical production.	MT is one of the research hotspots in the postharvest field in recent years.	[136,137,138]

## Data Availability

Not applicable.

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
