# Peer review of "Research Progress on Mango Post-Harvest Ripening Physiology and the Regulatory Technologies"

_foods, 2022, doi:10.3390/foods12010173_

Round 1

Reviewer 1 Report

This manuscript review on “Research progress on mango post-harvest ripening physiology and the regulatory technologies”. The review paper is not comprehensive for mango as there are many other studies on mango postharvest have been excluded from this manuscript. The manuscript needs some major corrections :

 Points for the authors to address:

11. This review paper is missing the postharvest pathogens aspect. Pathogen is an essential area to be included for this review.

22. Section 2.1  Changes in respiration, ethylene, and energy levels during post-harvest ripening of mango. This section should be elaborated by providing more review on respiration rate, ethylene production as well as the response to the external ethylene during the supply chains (storage, transport, etc). Also, how this important physiological behaviour would change at different temperature.

33. Section 3.1 Physical treatment …. This section should be elaborated by providing more review on physical treatments. For example, for heat treatment … there are more references are not review as comparison. For example of this paper is excluded from this manuscript in the review of heat treatment for mango : Ethylene synthesis in mango fruit following heat treatment by S. Ketsa, S. Chidtragool, J. D. Klein and S. Lurie (1999).  https://doi.org/10.1016/S0925-5214(98)00060-X

4 4. Another example  of in  Section 3.1 Physical treatment …., lights treatment … UV-C is excluded from this manuscript. There are many studies utilising UV-C for extending mango postharvest life.

5.  5. Section 3.2.3 … Edible coating …. Only 6 papers were reviews, while other edible coating are available. For example of paper excluded from this review  such as : Carboxymethyl chitosan-pullulan edible films enriched with galangal essential oil: Characterization and application in mango preservation by  W. Zhou, Y. He, F. Liu, L. Liao, X. Huang, R. Li, et al (2020) https://www.sciencedirect.com/science/article/pii/S0144861720317525

Author Response

Response Letter

Dear editor:

Thank you very much for giving us an opportunity to revise our manuscript entitled “Research progress on mango post-harvest ripening physiology and the regulatory technologies” (ID: foods-205646).

We have revised our manuscript carefully according to the comments, and would like to re-submit it for your consideration. The amendments are highlighted in red font in the revised manuscript. Our responses to the referees are listed by point to point in the following sections. If there are any deficiencies in this manuscript modification, we will fully cooperate to editor and revise it again in the next time.

We would like to express our great appreciation to you and reviewers for the comments and hope that revised manuscript is acceptable.

In addition, we would also like to thank you again for your patience. As our main authors have been infected with COVID-19 and are under home quarantine without office computers and key files, we have been delayed for a few days in returning the revised manuscript. Thank you again

Look forward to hearing from you.

Yours sincerely,

Dr. Bangdi Liu and Prof. Jing Sun

Academy of Agricultural Planning and Engineering, Ministry of Agriculture and Rural Affairs of the People's Republic of China

PO Box 111, No. 41 Maizidian Road, Beijing, 100083, China

E-mail: bangdi.liu@foxmail.com 

Reply to Referees

to Referees 1

Question1: This review paper is missing the postharvest pathogens aspect. Pathogen is an essential area to be included for this review.

Response: Thanks for your valuable suggestions. Indeed, pathogenic microorganisms, especially anthracnose, are an important factor affecting mango ripening during storage. We have added Section 2.5 to the manuscript to elaborate on the relationship between microorganisms infection and mango ripening.

Question2: Section 2.1 Changes in respiration, ethylene, and energy levels during post-harvest ripening of mango. This section should be elaborated by providing more review on respiration rate, ethylene production as well as the response to the external ethylene during the supply chains (storage, transport, etc). Also, how this important physiological behaviour would change at different temperature.

Response: Thanks for your valuable suggestions. We have added detailed explanation of response of mango ethylene and respiration to external ethylene in Section 2.1 as suggested by reviewers. In addition, we described in Section 2.1 the response of ethylene synthesis and respiration to temperature during mango ripening.

Question3: Section 3.1 Physical treatment …This section should be elaborated by providing more review on physical treatments. For example, for heat treatment … there are more references are not review as comparison. For example of this paper is excluded from this manuscript in the review of heat treatment for mango : Ethylene synthesis in mango fruit following heat treatment by S. Ketsa, S. Chidtragool, J. D. Klein and S. Lurie (1999).  https://doi.org/10.1016/S0925-5214(98)00060-X

Response: Thanks for your valuable suggestions. We have made major changes to Section 3.1. Especially in 3.1.5 and 3.1.6, those two parts were mainly rewritten. In Section 3.1.5, the reference provided by reviewers and other related references on the influence of heat treatment on mango ripening were added.

Question4: Another example  of in  Section 3.1 Physical treatment …., lights treatment … UV-C is excluded from this manuscript. There are many studies utilising UV-C for extending mango postharvest life.

Response: Thanks for your valuable suggestions. We have made major changes to Section 3.1. In Section 3.1.6,We changed the original title (light treatment) to irradiation treatment. In this part, the effects of ultraviolet light (UV), visible light and electron beam irradiation (E-beam) on mango preservation and ripening are described in detail.

Question5: Section 3.2.3 … Edible coating …. Only 6 papers were reviews, while other edible coating are available. For example of paper excluded from this review  such as : Carboxymethyl chitosan-pullulan edible films enriched with galangal essential oil: Characterization and application in mango preservation by  W. Zhou, Y. He, F. Liu, L. Liao, X. Huang, R. Li, et al (2020)

Response: Thanks for your valuable suggestions. We have added several references including the reference provided by reviewers on the effects of edible coating on mango ripening in Section 3.2.3.

Reviewer 2 Report

In the present manuscript, mango post-harvest changes and regulatory technologies were studied. It provides a clear and useful description of mango post-harvest physiology and regulatory issues, as well as well-constructed conclusions. The manuscript is relevant and novel with respect to the existing literature. Existing errors in the literature and figure presentations, as well as citing research without mentioning references, should be corrected.

Specific comments
1. Page 2: See text and Figure 2 as an example:

...As shown in Figure 2... Decomposition of acid or acids?...

2. If other research is mentioned, it should be supported by references: Page 7, 9, 13: Some studies...
(REF?); Page 6: Few studies... (REF?) ...

3. There are many errors in the references in the paper:

-incorrect (ref. 18-Ma et al.) -

-check reference - Hu et al. (15) Mango? Sweet Oranges? Han (19) Mango?

-duplication of incorrect references (54, 55)

4. Errors in Figure 4: for example, ioprene”, ..

Author Response

Response Letter

Dear editor:

Thank you very much for giving us an opportunity to revise our manuscript entitled “Research progress on mango post-harvest ripening physiology and the regulatory technologies” (ID: foods-205646).

We have revised our manuscript carefully according to the comments, and would like to re-submit it for your consideration. The amendments are highlighted in red font in the revised manuscript. Our responses to the referees are listed by point to point in the following sections. If there are any deficiencies in this manuscript modification, we will fully cooperate to editor and revise it again in the next time.

We would like to express our great appreciation to you and reviewers for the comments and hope that revised manuscript is acceptable.

In addition, we would also like to thank you again for your patience. As our main authors have been infected with COVID-19 and are under home quarantine without office computers and key files, we have been delayed for a few days in returning the revised manuscript. Thank you again

Look forward to hearing from you.

Yours sincerely,

Dr. Bangdi Liu and Prof. Jing Sun

Academy of Agricultural Planning and Engineering, Ministry of Agriculture and Rural Affairs of the People's Republic of China

PO Box 111, No. 41 Maizidian Road, Beijing, 100083, China

E-mail: bangdi.liu@foxmail.com

to Referees 2

Question1:Page 2:See text and Figure 2–as an example:

...As shown in Figure 2...Decomposition of acid or acids?...

Response: This is a mistake in our writing. We have checked Figure 2 in detail to correct the language errors

Question2:If other research is mentioned, it should be supported by references:Page 

7, 9, 13: Some studies... (REF?); Page 6: Few studies... (REF?)..

Response: Thanks for your valuable suggestions. We checked the full manuscript and added references to the parts of "numerous research points" and "some research points".

Question3:There are many errors in the references in the paper:

-incorrect (ref. 18-Ma et al.)

-check reference - Hu et al. (15) Mango? Sweet Oranges? Han (19) Mango?

-duplication of incorrect references (54, 55)

Response: Thank you for your valuable comments. We carefully check several references pointed out by the reviewer and revised the description

Question 4:Errors in Figure 4: for example, ioprene”, ..

Response: We have revised in Figure 4, it was typo.

Reviewer 3 Report

The manuscript ' Research progress on mango post-harvest ripening physiology and the regulatory technologies by Liu et al., summarized the effect of ripening physiology of mangoes and technologies to improve postharvest life of mango fruits. On the basis of the reviews, authors concluded that after harvest physiological changes in mango are  complex process, involving ethylene re-lease, respiration, energy supply, sugar metabolism, acid degradation, terpene volatile synthesis, carotenoid accumulation, and degradation of cell wall pectin. Similarly for regulation of ripening various techniquies including physical, chemical preservatives, and en-dogenous biological hormones can be used. This topic is suitable for the journal readers and the authors gathered a helpful and interesting information.  I suggest to include some more references on effect of ionizing radiationg on storage life of mango fruits. For further improvement of review, a summarized table involving use of chemical treatment and waxes for enhancing storage life of mango may be incorporated.  

Author Response

Response Letter

Dear editor:

Thank you very much for giving us an opportunity to revise our manuscript entitled “Research progress on mango post-harvest ripening physiology and the regulatory technologies” (ID: foods-205646).

We have revised our manuscript carefully according to the comments, and would like to re-submit it for your consideration. The amendments are highlighted in red font in the revised manuscript. Our responses to the referees are listed by point to point in the following sections. If there are any deficiencies in this manuscript modification, we will fully cooperate to editor and revise it again in the next time.

We would like to express our great appreciation to you and reviewers for the comments and hope that revised manuscript is acceptable.

In addition, we would also like to thank you again for your patience. As our main authors have been infected with COVID-19 and are under home quarantine without office computers and key files, we have been delayed for a few days in returning the revised manuscript. Thank you again

Look forward to hearing from you.

Yours sincerely,

Dr. Bangdi Liu and Prof. Jing Sun

Academy of Agricultural Planning and Engineering, Ministry of Agriculture and Rural Affairs of the People's Republic of China

PO Box 111, No. 41 Maizidian Road, Beijing, 100083, China

E-mail: bangdi.liu@foxmail.com 

to Referees 3

Question1:I suggest to include some more references on effect of ionizing radiationg on storage life of mango fruits. For further improvement of review, a summarized table involving use of chemical treatment and waxes for enhancing storage life of mango may be incorporated.

Response: This question is the same as Question 4 of Reviewer 1. Thanks for your valuable suggestions. We have made major changes to Section 3.1. In Section 3.1.6,We changed the original title (light treatment) to irradiation treatment. In this part, the effects of ultraviolet light (UV), visible light and electron beam irradiation (E-beam) on mango preservation and ripening are described in detail.

Round 2

Reviewer 1 Report

This manuscript has been revised according to the recommendations. However, the revised paper hasn’t comprehensive yet as a review paper as there are many other studies on mango postharvest have been excluded from this manuscript. For example, there is only one paper of  UV-C treatment and one paper of UV-B treatment on Mango are  included in this paper. Here is the list of examples of paper on mangos that have been excluded from this manuscript;

·        Milton Vega-Alvarez, et al (2020). Metabolomic Changes in Mango Fruit Peel Associated with Chilling Injury Tolerance Induced by Quarantine Hot Water Treatment.

·        Dionisio G. Alvindia and Miriam A. Acda (2015).  Revisiting the efficacy of hot water treatment in managing anthracnose and stem-end rot diseases of mango cv. ‘Carabao’

·        Yilma Dessalegn, et al (2013). Integrating plant defense inducing chemical, inorganic salt and hot water treatments for the management of postharvest mango anthracnose.

·        Dov Prusky , et al (1989). Effect of hot water brushing, prochloraz treatment and waxing on the incidence of black spot decay caused by Alternaria alternata in mango fruits.

·        G.A. González-Aguilar et al, (2007). Improving postharvest quality of mango ‘Haden’ by UV-C treatment.

·        Xiaokai Li, et al (2022). Effect of Enhanced Ultraviolet-B Radiation on Fruit Maturity and Quality and Leaf Photosynthesis in ‘Guifei’ Mango.

 Therefore, the revised manuscript needs some major corrections.

Author Response

Response Letter

Dear editor and reviewer:

Thank you very much for giving us an opportunity to revise our manuscript entitled “Research progress on mango post-harvest ripening physiology and the regulatory technologies” (ID: foods-205646).

We have revised our manuscript carefully according to the comments, and would like to re-submit it for your consideration. The amendments are highlighted in red font in the revised manuscript. Our responses to the referees are listed by point to point in the following sections. If there are any deficiencies in this manuscript modification, we will fully cooperate to editor and revise it again in the next time.

We would like to express our great appreciation to you and reviewers for the comments and hope that revised manuscript is acceptable.

Look forward to hearing from you.

Yours sincerely,

Dr. Bangdi Liu and Prof. Jing Sun

Academy of Agricultural Planning and Engineering, Ministry of Agriculture and Rural Affairs of the People's Republic of China

PO Box 111, No. 41 Maizidian Road, Beijing, 100083, China

E-mail: bangdi.liu@foxmail.com 

Reply to Referees

to Referees 1

Question:

This manuscript has been revised according to the recommendations. However, the revised paper hasn’t comprehensive yet as a review paper as there are many other studies on mango postharvest have been excluded from this manuscript. For example, there is only one paper of UV-C treatment and one paper of UV-B treatment on Mango are included in this paper. Here is the list of examples of paper on mangos that have been excluded from this manuscript;

Milton Vega-Alvarez, et al (2020). Metabolomic Changes in Mango Fruit Peel Associated with Chilling Injury Tolerance Induced by Quarantine Hot Water Treatment.

Dionisio G. Alvindia and Miriam A. Acda (2015).Revisiting the efficacy of hot water treatment in managing anthracnose and stem-end rot diseases of mango cv. ‘Carabao’

Yilma Dessalegn, et al (2013). Integrating plant defense inducing chemical, inorganic salt and hot water treatments for the management of postharvest mango anthracnose.

Dov Prusky , et al (1989). Effect of hot water brushing, prochloraz treatment and waxing on the incidence of black spot decay caused by Alternaria alternata in mango fruits.

G.A. González-Aguilar et al, (2007). Improving postharvest quality of mango ‘Haden’ by UV-C treatment.

Xiaokai Li, et al (2022). Effect of Enhanced Ultraviolet-B Radiation on Fruit Maturity and Quality and Leaf Photosynthesis in ‘Guifei’ Mango.

Therefore, the revised manuscript needs some major corrections.

Response: 

Thank you for your confirmation of our last round of revisions, and thank you for your new suggestions.

We have already added several references provided by reviewers and other relevant references found on web of Science Direct. However, the reference of Xiaokai Li, et al (2022). Effect of Enhanced Ultraviolet-B Radiation on Fruit Maturity and Quality and Leaf Photosynthesis in ‘Guifei’ Mango would not be cited. Because this is a paper on the effects of UV-B on the growth and ripening of mango during pre-harvest planting.

In addition, I would like to state that the main review purpose in this paper is the physiological changes and ripening regulation of mango during postharvest storage. Therefore, there are some technical studies on mango storage quality that are not within the scope of our review in this paper. For example, many studies on mango chilling injury and infectious diseases have been reviewed in specialized plant-pathology and microbiology papers. Hence, some articles on mango postharvest technology without ripening would not be introduced into this review. This is why reviewers will find some references on the website that is excluded from our review.

However, we also humbly accept the suggestions put forward by the reviewers and are willing to continue to cooperate with the revision in the future.
